



# Using Sentinel-1 wet snow maps to inform fully-distributed physically-based snowpack models

Bertrand Cluzet[1], Jan Magnusson[1], Louis Quéno[1], Giulia Mazzotti[2,1], Rebecca Mott[1], and Tobias Jonas[1]

[1]WSL Institute for Snow and Avalanche Research SLF, Davos, Switzerland
[2]Univ. Grenoble Alpes, Université de Toulouse, Météo-France, CNRS, CNRM, Centre d'Études de la Neige, Grenoble, France

**Correspondence:** Bertrand Cluzet (bertrand.cluzet@slf.ch)

**Abstract.** Distributed energy and mass-balance snowpack models at sub-kilometric scale have emerged as a tool for snow-hydrological forecasting over large areas. However, their development and evaluation often rely on a handful of well observed sites on flat terrain with limited topographic representativeness. Validation of such models over large scales in rugged terrain is therefore necessary. Remote sensing of wet snow has always been motivated by its potential utility in snow hydrology.

However, its concrete potential to enhance physically based operational snowpack models in real time remains unproven. Wet snow maps could potentially help refining the temporal accuracy of simulated snowmelt onset, while the information content of remotely sensed snow cover fraction pertains predominantly to the ablation season. In this work, wet snow maps, derived from Sentinel-1 and snow cover fraction (SCF) retrieval from Sentinel-2 are compared against model results from a fully distributed energy-balance snow model (FSM2oshd). The comparative analysis spans the winter seasons from 2017 to 2021, focusing on

the geographic region of Switzerland. We use the concept of wet snow line (WSL) to compare Sentinel-1 wet snow maps with simulations. We show that while the match of the model with flat-field snow depth observation is excellent, the WSL reveals insufficient snow melt in the southern aspects. Amending the albedo parametrization within FSM2oshd allowed achieving earlier melt in such aspects preferentially, thereby reducing WSL biases. Biases with respect to Sentinel-2 snow line (SL) observations were also substantially reduced. These results suggest that wet snow maps contain valuable real-time information

for snowpack models, nicely complementing flat-field snow depth observations, particularly in complex terrain and at higher elevations. The persisting correlation between wet snow line and snow line biases provides insights into refined development, tuning and data assimilation methodologies for operational snow-hydrological modelling.





## 1 Introduction

Seasonal snow is crucial for the hydrology of mountains and their tributaries as it stores wintertime precipitation and entails
delayed runoff (Barnett et al., 2005). Snowmelt runoff is key for downstream agriculture (Qin et al., 2020) and hydropower
production (Gaudard et al., 2014). Intense snowmelt can also lead to dramatic floods (Diffenbaugh et al., 2013). As the inter-
annual variability of snow distribution is high, anticipating the amount and timing of melt is therefore essential, especially in a
changing climate (Lafaysse et al., 2014).

Snow models have therefore become indispensable in the field of hydrological forecasting (e.g. Griessinger et al., 2019).
However, accurately resolving all intricate processes that influence the mass and energy balance of the snowpack across var-
ious spatial scales is difficult. The accumulation of snow in a region is contingent upon precipitation patterns, which exhibit
substantial variability both regionally (Isotta et al., 2014) and locally (Scipión et al., 2013). Wind and gravitational redistribu-
tion contribute to the spatial heterogeneity of snowpacks (Liston and Sturm, 1998; Freudiger et al., 2017; Mott et al., 2018).
Meanwhile, the surface energy balance, which governs the amount of snowmelt, is subject to fine-scale variations. These
variations are influenced by topographical features and cloud cover, which impact incoming shortwave and longwave radia-
tion (Brauchli et al., 2017; Jonas et al., 2020) as well as air temperature. Wind patterns, too, can exert influence on sensible
and latent heat fluxes at small scales (Mott et al., 2011). In summary, the accurate modelling of snowpack ablation necessitates
the comprehensive representation of both the variability in snow cover and the multifaceted drivers affecting its energy balance.

Snow melt is not equivalent to snowmelt runoff. Snowmelt runoff can only commence when meltwater exits the snowpack
from its bottom. This journey entails complex processes related to liquid water storage and routing, introducing a substantial
temporal lag between the initiation of melt and eventual snowmelt runoff (Lundquist and Dettinger, 2005). Towards the end
of the accumulation season, an increase of the surface energy input typically gives rise to more frequent and intense episodes
of surface melt, a phase often referred to as the moistening phase. Concurrently, rainfall events may further augment the liq-
uid water content at the surface. Once the upper layers of the snowpack reach saturation, liquid water gradually infiltrates
through the snowpack during the ripening phase. This drainage process is influenced by various factors, including refreezing,
the presence of preferential flow paths, and capillary barriers (Wever et al., 2014; Würzer et al., 2017), as well as the occurrence
of ice layers (Quéno et al., 2020). Ultimately, snowmelt runoff commences when the liquid water exits the snowpack at its base.

In the context of large scale and high spatial resolution snowmelt applications, compromises are needed to find a good
trade-off between model accuracy and computational costs. Often, intermediate-complexity snowpack models provide a good
balance (Magnusson et al., 2015). These models solve the energy and mass balance of the snowpack with simplifications as to
the representation of the internal snowpack structure (Marsh et al., 2020; Essery, 2015). Particularly, they use a limited number
of numerical layers (Cristea et al., 2022) resulting in a simplified description of liquid water routing and snow metamorphism.
These factors, combined with further uncertainties in meteorological forcings, may result in considerable modelling errors



(Raleigh et al., 2015; Krinner et al., 2018), which are often hard to assess because of an intrinsic lack of observational data in mountainous areas.

Satellite observations of the snowpack could help to constrain such model errors, provided that they have the adequate spatio-temporal resolution and sufficient accuracy (Largeron et al., 2020). Snow cover fraction (SCF) observations from Sentinel-2, for instance, are available at a resolution of 20m every three to five days, however only in the absence of clouds. The C-band synthetic aperture radar (SAR) sensor on Sentinel-1, on the other hand, is sensitive to the presence of liquid water in the snowpack irrespective of cloud cover (Veyssière et al., 2019; Marin et al., 2020; Karbou et al., 2021). Binary wet snow maps (WSM)
can be derived from SAR backscatter at a resolution higher than 100m, requiring only small amounts of liquid water for snow to be classified as wet (Nagler et al., 2016).

The timing of information is key in real-time operational modelling. Snow cover fraction and wet snow maps provide information at different times during the season. SCF observations provide information about the first snowfall, and during the
period of partial snow cover during the ablation season. This information can be used to reconstruct snow water equivalent (Margulis et al., 2016), but is not helpful for determining the onset of snowmelt in real-time (Andreadis and Lettenmaier, 2006; Buchelt et al., 2022). Wet snow retrievals, on the other hand, provide qualitative information during the onset of snowmelt, leaving the entire ripening and ablation phase to exploit the information. Key to operational snow-hydrological model applications, wet snow retrievals can anticipate the onset of snowmelt, making them a potentially interesting data source for assimilation
(Malnes et al., 2015).

Nevertheless, several questions remain as to the potential utility of wet snow maps in this context. Firstly, the SAR signal may capture superficial short-lived melt events which are very difficult to capture by intermediate-complexity snowpack models. These models do not capture the intricate microstructure and stratigraphy of the snow, thereby complicating the assimilation
process. The duration of the ripening phase, associated with the delay between the SAR signal and the onset of snowmelt runoff, is also arguably very variable, depending on the snowpack thickness, internal properties, and springtime weather conditions (Marin et al., 2020). Finally, due to the sensitivity of radar retrievals to seasonal change in land surface properties (Karbou et al., 2022) and the unavailability of meaningful retrievals in forested areas, reliable observations are confined to elevations above the treeline and away from water bodies and human activities.

In this work, we assess whether information from wet snow maps can be used to inform a snow hydrological model used in nationwide operational forecasting in practice. We use two versions of our physically-based multi-layer snowpack model FSM2oshd (Mott et al., 2023), which we run operationally over entire Switzerland at 250m spatial resolution. These two versions have different fresh snow albedo parametrizations. Snow depth data from nearly 450 monitoring stations were assim-
ilated into these simulations, to ensure a good model performance free of systematic overall biases to start with. Note that this approach was chosen to reveal the value of information contained in WSM and SCF products beyond the identification and



removal of obvious model biases. The study focuses on five ablation seasons (March-July) from 2017 to 2021 addressing the following questions:

- Is information from Sentinel-2 snow cover fraction and Sentinel-1 wet snow complementary to already available flat field snow depths?

- Is it possible to leverage remotely sensed data on wet snow to enhance the accuracy of snowpack simulations in complex terrain?

The study area, observations and modelling chain are described in Sec. 2. The results are presented and discussed in Sec. 3 & 4. We finally conclude and open research perspectives in Sec. 5.

## 2 Material and methods

### 2.1 Study area

The study area encompasses Switzerland and hydrologically connected areas outside its boundaries (see Fig. 1). The domain size stretches over 272km in the latitudinal direction and 365km in the longitudinal direction, totalling 920,884 model grid cells at a resolution of 250m. We focus our analysis on 16 hydrological units (MEZ), which cover the alpine area of the domain. The average size of the MEZs is about 1600km$^2$, ranging from 721km$^2$ (MEZ 16) and 4470km$^2$ (MEZ 9). All of these units are located within Switzerland, some of them extending to neighbouring countries (Liechtenstein, Austria, Germany, Italy, and France). The MEZs are situated on both sides of the main alpine ridge and encompass diverse hydro-climatic conditions. There are 444 locations with daily in-situ snow depth observations available throughout the domain indicated by the red dots in Figure 1. Among these observations, 255 are within the considered MEZs.

### 2.2 Satellite observations from AlpSnow

AlpSnow is a project from the European Spatial Agency (ESA) which provides a 5-year demonstration dataset for novel remote-sensing retrievals of snow parameters over the Alps. In this paper, we use a wet snow maps product, retrieved from Sentinel-1 and snow cover fraction from Sentinel-2.

#### 2.2.1 Wet Snow

Wet snow maps from AlpSnow are processed from Sentinel-1 C-band synthetic aperture radar backscatter time series. C-band backscatter is sensitive to small amounts of liquid water in the snowpack, causing a sharp decrease in backscatter compared to bare ground or dry snow conditions (Mätzler, 1987). A -2.5dB threshold with respect to a reference scene is applied to detect wet snow, resulting in binary wet snow maps (Nagler et al., 2016). As evidenced by Marin et al. (2020), the backscatter signal resumes to dry snow conditions as the snowpack becomes patchy throughout the ablation season, even though at that time the snowpack is usually fully saturated with liquid water. Therefore, melting, patchy snow cannot be distinguished from dry



snow or bare ground. Furthermore, due to the viewing slant angle, geometric distortions (shadows, layover and foreshortening) cause data gaps in rugged terrain. Satellite acquisition occurs around 5UTC (descending orbit) or 17UTC (ascending orbit) with a revisit time of 3 to 6 days during the study period. AlpSnow wet snow maps resolution is $0.001^o$x$0.001^o$ in WGS84 (EPSG:4326) corresponding to approximately 100mx100m in the modelling coordinate system (EPSG:21781).

120

Wet snow maps were aggregated into pixels of 250m coinciding with the model grid. Products were first reprojected and resampled at 50m using a nearest neighbour approach into the modelling coordinate system. Then, these products were masked using a 10m forest mask (Mott et al., 2023). Urban areas were also excluded using class 50 of the Copernicus Land Cover Mask (Buchhorn, 2022). For each 250m pixel, an aggregated binary value was computed as follows:

- If more than 90% of the 50m-subpixels were forest/urban, then the pixel was masked as forest/urban.

- Else, if more than 5% of the 50m-subpixels were classified as geometric distortions, then the pixel was masked out.

- Otherwise, the 250m wet snow binary value was equal to the majority of the valid subpixel values.

Fig. 2a and b show that due to overlapping tracks, 5 to 10 Sentinel-1 wet snow observations were available per month. Some areas (MEZ 6, 14, 17 and 22) are even observed ten times a month in the morning and evening.

**2.2.2 Snow Cover Fraction**

AlpSnow snow cover fraction is derived by spectral unmixing on Sentinel-2 level 1C radiance at the pixel level (Keuris et al., 2023). Product resolution is $0.0002^o$x$0.0002^o$ in WGS84 (EPSG:4326), (approximately 20mx20m in EPSG:21781). Satellite acquisition time is around 10–11h UTC. The following Sentinel tiles were processed: 'T31TGM', 'T32TMT', 'T32TNT', 'T32TLS', 'T32TMS', 'T32TNS'. Based on optical data, these products are inherently limited by the following factors (e.g. Barrou Dumont et al., 2021; Keuris et al., 2023):

- cloud/snow discrimination, cloud shadows

- missed detection of cirrus clouds

- shaded snow in steep terrain and sparse forests, from November to March,

- non-zero snow cover fraction attributed to snow-free glaciers, usually in Summer.

We mitigate these limitations by focusing on observations available in open terrain, in the ablation season from April to June.

SCF images were aggregated following a similar procedure as for the WSM (see Sec. 2.2.1). Products were reprojected and and resampled at 10m. For each 250m pixel, an aggregated value was computed as follows:

- if more than 90% of the 10m-subpixels were forest/urban, then the pixel was masked as forest/urban.





– else, if more than 5% of the 10m-subpixels were covered by clouds, the the pixel was masked as cloudy.

   – otherwise, the 250m SCF value was computed as the average value of the valid (non forested, urban or cloudy) subpixels.

Fig. 2c shows the average value of valid Sentinel-2 SCF observations per pixel during the spring period from April to June (thereby accounting for cloud occurrence). Due to overlapping tracks, the central portion of the domain, covering MEZ 6, 11, 18, 20 and 22 has twice as many Sentinel-2 overpasses as the rest of the domain. On average, 2 to 5 cloud-free observations
are available per pixel each month.

## 2.3   FSM2oshd

In this work, we use the physically-based multi-layer snowpack model FSM2oshd (Mott et al., 2023), which we run operationally over entire Switzerland at 250m spatial resolution. FSM2oshd includes an optimal-interpolation based data assimilation scheme, using in-situ snow depth measurements from 444 stations (see Fig. 1) to correct for snowfall input errors
(Magnusson et al., 2014). To account for unrepresented preferential deposition, wind transport, and gravitational redistribution of snow, a terrain-dependent correction is applied to solid precipitation based on Vögeli et al. (2016). In its standard configuration, FSM2oshd is a 3-layer snowpack model where liquid water routing through the snowpack is modelled using a bucket approach. A subgrid parametrization is used to derive pixel-level snow cover fraction from seasonal values of SWE and HS, accounting for subpixel terrain roughness and recent snowfall history (Helbig et al., 2021). This is an essential component
when it comes to comparing model output with satellite retrievals of snow cover fraction. The snowpack model is run using output from the numerical weather prediction system COSMO-1E operated by MeteoSwiss, which was carefully debiased and downscaled to 250m using a range of dynamical and statistical methods. More details about FSM2oshd are available in Mott et al. (2023), describing in-depth the entire modelling framework including preprocessing of input data and available data assimilation schemes.


Here, we use two versions of FSM2oshd: a) the pre-operational version fsm_preop was configured to match snow depth observations from several Swiss monitoring networks, mostly at flat unforested measuring sites. fsm_preop, however, was not tuned in any way using spatially fully-distributed datasets, such as remote sensing products; b) the version fsm_optim was optimized to address a model deficiency detected in comparison with Sentinel-1 wet snow retrievals (c.f. Sec. 3.2). The
optimization consisted of lowering the fresh snow albedo from 0.92 in fsm_preop to 0.86 in fsm_optim, to accelerate the onset of snowmelt particularly in south facing aspects.

## 2.4   Comparing the model with observations

In order to compare the simulations with observations, model outputs were extracted and masked out when the forest fraction exceeded 80% of the grid cell. The same mask was applied to the observations, to avoid strong geospatial bias in the analysis.
The definition of a "wet snow" state for the model is required for a direct comparison with the observations. Given the high sensitivity of the observation to small liquid water amounts in the snowpack, we considered modelled snow to be wet as soon





as liquid water was present in any of the snow layers at the time of the acquisition (either 05UTC or 17UTC).

## 2.5 Deriving the Wet Snow line

The melting season usually starts with a progressive moistening of the dry snowpack from low to high elevations. The wet snow line (WSL) is the time-varying elevation of this wetting front (Gupta, 2011), which primarily depends on topographic factors. As shown by Karbou et al. (2021, 2022), wet snow maps from Sentinel-1 capture this phenomenon well. The WSL is conceptually very similar to the snow line (SL), which can be defined as the elevation where the snow cover fraction equals 50% (e.g. Parajka et al., 2010).


The WSL was estimated from Sentinel-1 wet snow maps and "wet snow" states from model outputs (Sec. 2.4) using the following algorithm. Using the MEZs as spatial aggregation units, we computed the WSL by aspect sectors each time a Sentinel-1 image was available. Pixels with a slope $< 8^o$ were considered as flat, and pixels above $30^o$ of slope were excluded from the aggregation. Only data between 1600 and 3200ma.s.l. were evaluated, to enable a robust estimation of WSL unaffected by

forests and high-elevation glacierized areas (see Sec. 2.2.1). All valid pixels in a given aspect were stratified by elevation. We then computed a weighted interpolation of the pixel values using a window width of 300m giving us the proportion of wet pixels (Wet Snow Fraction, $WSF$) as a function of the elevation as shown in Fig. 3.

For the observations, starting from dry conditions at high elevations we identify the maximum $WSF_{max}$ as we go to lower

elevations. If $WSF_{max}$ did not exceed 0.6, then the snowpack was not substantially wet, and no snow line was determined for that aspect class. Otherwise, the WSL was defined as the lowest elevation above that of $WSF_{max}$ with a $WSF <= 0.8 \times WSF_{max}$. If this value was not reached, the snow cover was assumed to be wet over the entire elevational range considered, and no WSL was determined either.

The snow line (SL) was computed using the same weighted interpolation to arrive at an elevational profile of SCF. The SL

was then defined as the highest elevation with $SCF < 0.5$.

## 3 Results

### 3.1 Performance of the snow model against flat-field observations

Fig. 4 shows the performance of both FSM versions, fsm_preop and fsm_optim against flat field observations between 2000

and 2500 ma.s.l. across Switzerland for the five considered years. Overall, the match is excellent, in particular during the accumulation period, which is consistent with the fact that these observations were assimilated for correcting errors in the snowfall input (see Sec. 2.3). For both model versions, the most notable differences with respect to the observations arise in the early





ablation period, in particular during spring 2019 and 2021. As expected, due to a slightly lower fresh snow albedo, snow melt is initiated slightly earlier in fsm_optim compared to fsm_preop. Aimed mostly at south facing slopes, the optimisation had
only limited influence on simulations at our flat field monitoring sites. The accelerated melt, however, leads to slightly stronger negative biases, albeit within acceptable limits. This is particularly notable in seasons when fsm_preop already demonstrates a slight negative bias, such as in the winter years of 2019 and 2021.

## 3.2 Spatial comparison and topographic aggregation of satellite observations

Satellite data provide much more information on the spatial performance of the model than the station dataset. As an example, Fig. 5 shows the wet snow map aggregated to the model resolution of 250m compared with equivalent model simulations with fsm_preop for MEZ 11 on 1st April 2019. Both simulated and observed maps are dry at high elevations, and transition into wet conditions at lower elevations. In the south slopes, the observations indicate wet conditions up to higher elevations than the model simulation. No retrieval is available in some steep slopes of the South West aspect due to geometric distortions
(shadowing) of the SAR signal.

In order to gain quantitative insights into the spatial dynamics of wet snow in both observations and model, map products were aggregated by slope classes in Fig. 6, including respective evaluations of the wet snow line. Observations (Fig. 6a) and fsm_preop (Fig. 6b) show good agreement in the northern sectors as well as in flat terrain. However, in the South-West to East
slopes, the simulated WSL consistently shows notably lower values compared to the observations (as also observed in Fig. 5). This suggests a delayed initiation of simulated snow melt in sun-exposed terrain. This finding led to testing whether lowering the fresh snow albedo would mitigate existing model biases by enabling an earlier onset of melt preferentially in sun exposed slopes, motivating the parameter adaptation applied in fsm_optim. Indeed, simulations of the wetting front with fsm_optim (Fig. 6c) show a substantially better match to observations, particularly in West and South aspects, but also for flat terrain, with
a raise in the WSL by up to 300m relative to the baseline simulations with fsm_preop.

## 3.3 Time series of snow line and wet snow line

Generalizing the approach, both WSL and SL were computed over the course of the melt season, for all MEZs, and all years. Results for MEZ 11 are shown in Fig. 7. Using fsm_preop, we see that the wet snow line gradually rises to high elevations from March to May each year, both in the observations and in the model (Fig. 7, left). From June onwards (and even May in
2018 and 2020), the snowpack is wet up to high elevations and no WSL can be defined, i.e. the signal is saturated both in the observations and the model. In the North sector, the match between observations and simulations is good, except in 2017 where the model shows a considerably delayed onset of melt for elevations between 2200m and 2400m. Southerly exposures, characterized by significant inter-annual variability, the model consistently exhibits a prolonged delay throughout the season. The simulated wet snow tends to be frequently positioned about 300m lower than observed values. From April onwards, fsm_optim
shows a higher WSL than the fsm_preop version, usually by 200 to 300m. The match with observations is therefore better, in





particular in the South exposed slopes.

Compared to the assessment of the WSL, the SL time series show a much better agreement between observations and simu-lations than the WSL (Fig. 7, right). The agreement between observations and fsm_preop is particularly good in the Northerly
slopes and in flat terrain, for almost all years. However some expositions show a too high snow line for the model (e.g. flat in 2018), and the snow line is most of the times too low in the South sector (years 2017, 2020 and 2021 in particular). The SL of fsm_optim becomes significantly higher (100–200m) than that of fsm_preop from mid-May onwards. This results in an overall better match with the observations, with some exceptions (flat exposition in 2018 for example).

Comparing both WSL and SL products, we note that the years with the lowest SL bias for fsm_preop seem to coincide with the strongest WSL biases for that model version. Moreover, for a given elevation, there is a fairly consistent time lag between the WSL and the SL of one to two months.

### 3.4 Relationship between spring seasonal biases of WSL and SL

The time series of Fig. 7 show a persistence of WSL biases in fsm_preop across the entire season, suggesting a potential cor-relation with SL biases. In essence, our hypothesis posits that a delayed onset of melt in the model will likely lead to a delayed melt-out. To confirm this hypothesis, we conducted an analysis by computing the average model WSL bias and SL bias per year (over March-May and April-June, respectively), aspect and MEZ. Fig. 8a shows results for fsm_preop for all five years, all aspects, and MEZ 11. Overall, some correlation ($R^2 = 0.32$) can be observed between the WSL and the SL biases, where often
fsm_preop is too dry coinciding with overestimated snow cover extent. WSL biases, however, are much higher in amplitude than SL biases (around -500–0m, vs -200–100m respectively). The biases vary both between years and aspect classes. For instance, 2021 exhibits overall strong biases, while 2018 is almost unbiased. The year 2019 stands out from the other years featuring a slightly higher SL bias. This anomaly is likely due to a single fresh snow event in May during which the snow line was several hundred meters too high in the model (Fig. 7). Within each year, the WSL bias for fsm_preop is dominated
by a North-South contrast. While the northern slopes (upwards pointing triangle) typically exhibit the least negative biases, southern slopes (downwards pointing triangle) exhibit the most negative biases, in the range of -500m to -200m. Similarly, the SL biases feature a North-East (square) to South-West (pentagram) contrast. This results in a very specific pattern: for each year, as we circle through the expositions clockwise, the bias values draw an ellipse anti-clockwise.

The predominant factor contributing to the North-South disparity in ablation is likely the shortwave radiative input. Expect-edly, a lower fresh snow albedo value, as implemented in fsm_optim (see Sec. 2.3), could therefore induce an earlier melt in the model, while at the same time enhancing differential melting dynamics between North and South faces. Results with fsm_optim show considerably reduced biases (Fig. 8b). Overall, with fsm_optim biases are better centred around zero, while the range of biases between aspects for any given year are reduced, particular with regards to WSL biases. The correlation





between biases in WSL and SL is also reduced from 0.32 to 0.13. Yet, the overall biases continue to vary between years. For
instance, fsm_optim under-predicts both WSL and SL in 2021, while predominantly over-predicting WSL and SL in the year
2018. These results also stand when considering all the years and MEZ (Fig. 9). Despite the noticeable fluctuations between
years, it is noteworthy that biases in WSL are considerably lower with fsm_optim compared to fsm_preop. This reduction also
aligns with a significant decrease in the negative bias observed in SL.


## 4    Discussion

### 4.1    On the proposed wet snow line metric

In this study, we have evaluated the potential of Sentinel-1 based wet snow maps for the performance assessment of snow-
hydrological models in the ablation period. Leveraging the wet snow line concept facilitated a robust analysis of the infor-
mation provided by Sentinel-1 wet snow maps for high spatial resolution snowpack modelling. The topographic aggregation
facilitated the direct comparison between remotely sensed observations and model simulations, while achieving a good level of
robustness against individual observation errors and data gaps as done in the literature (Mary et al., 2013; Karbou et al., 2022).
Topographic aggregation makes the analysis more robust to missing or erroneous data in the observations, which is particularly
important in the context of data assimilation (Cluzet et al., 2021). At the same time, key spatial information were maintained
by resolving biases in terms of region, elevation, aspect, and slope, which has been shown to be of critical relevance (Cluzet
et al., 2020).

Figs. 3 and 6, as well as results from Karbou et al. (2021) suggest that this topographic aggregation can be further exploited to
address shortcomings of the WSL retrievals (see Sec. 2.5 and Fig. 3): if a snowpack is wet at a given altitude, it will be wet at
a lower altitude, all other things being equal. The wet snow line is therefore a condensed and potentially useful metric for both
remote sensing and modelling, while the direct assimilation of binary WSL or SL products into physically based snow models
is typically seen as unfavourable (Baba et al., 2018; De Lannoy et al., 2012).

The wet snow line analysis evidenced a systematic dry bias in the pre-operational version of our model, fsm_preop (Fig.
8a). The adjusted albedo in fsm_optim resulted in better simulations of snow cover fraction (Figs. 8b and 9b). In addition, the
slightly reduced variability of the WSL and SL biases (Figs. 8b and 9b) give us additional confidence in the correction of fresh
snow albedo. This proves the potential of such metric to inform snowpack models.

Relying solely on snow cover fraction observations would have made it challenging to point out the details of the model
deficiencies present in fsm_preop. Remaining bias patterns of fsm_optim (Figs. 8b, 9b and 9b) could be due to many reasons,
e.g. biases in the sub-grid parametrization for snow cover fraction, which derives modelled SCF from modelled snow mass
(Helbig et al., 2021), biases in the observations (e.g. Aalstad et al., 2020), or biases in either simulated snow accumulation or



melt. In this study, wet snow observations helped to disentangle the causation by pointing at a delayed melt, particularly in South-facing slopes. Further work can now focus on revealing other factors which contribute to the remaining snow line biases.

### 4.2 Potential for data assimilation and model development

Significant biases, reaching up to approximately -350m, persist in the modelled wet snow line for fsm_optim, (Fig. 8b). However, the extent to which additional information can be extracted from the observations remains uncertain. Observations are sensitive to wet snow processes occurring at the sub-grid scale which are not accounted for in the model. The model itself has a coarse vertical resolution with 3 snow layers, which may not be enough to capture small surface melt events detected by SAR (Cristea et al., 2022; Lund et al., 2022). Further, the bucket approach used to route the liquid water through the snowpack is

known to generate a slightly delayed onset of snow-melt runoff (Wever et al., 2014). On the other hand, wet snow retrievals are not flawless either. SAR is sensitive to even very small amounts of liquid water and may misinterpret wet soil below a dry snowpack as wet snow. The retrievals are further known to be problematic in the presence of fractional snow cover, and often report dry snow conditions towards melt out when this is clearly not the case (c.f. Sec 2.2). All these factors may contribute to an irreducible mismatch between observed and simulated wet snow maps, which may question the utility of wet snow maps

beyond model assessment and optimization as demonstrated in this study.

Several factors, however, suggest that wet snow maps contain further information, valuable for model evaluation and data assimilation in an operational model setting. First, it is interesting to note that even after model tuning, inter-annual variability dominates WSL and SL bias patterns, and that their magnitudes show a strong correlation (Fig. 8). For example, the year 2018

exhibits positive biases of WSL and SL, while it is the opposite in 2021. Operational snowpack simulations are intrinsically dependent on numerical weather prediction models, which show variable performance and systematic biases between years (Winstral et al., 2019). In one year, a slight error in temperature might turn a major snowfall event into rain while in another early season snowfall might be over-predicted, both causing snow model biases that will persist throughout the entire season. Such biases could be mitigated in real-time through assimilation of wet snow retrievals, since the transition from dry to wet

snow happens during the season, not at the end of the season when optical remote sensing reports melt-out.

The potential for data assimilation of SCF has long been established, although most of the time in a reanalysis context (Margulis et al., 2015; Aalstad et al., 2018; Alonso-González et al., 2021) rather than in real time (Baba et al., 2018). As long as the snow cover fraction equals 1, the only information that can be derived is that the snow water equivalent is above a few tens

of millimetres (e.g. Magand et al., 2014). This is not very informative as long as the model is not completely off from reality (De Lannoy et al., 2012). Fig. 7, however, frequently illustrates a persistent bias in the snow line throughout the ablation period. This pattern implies that information collected early at lower elevations, could potentially be transferred to higher elevations through data assimilation using the propagation of information (Cluzet et al., 2021; Alonso-González et al., 2023). Such an approach has the potential to enhance the overall performance of the model.

However, this may not be the case in the absence of systematic model biases. Fig. 7 shows that once the systematic albedo





bias has been fixed in fsm_optim, there is no longer a consistent snow line bias throughout the season: there is a negative snow line bias early in the season, and then a positive one later in the season. In such a context, while reanalyses may be able to find optimal trajectories once the full time series of observations has been gathered, real-time data assimilation might result in degraded performance in the late season. SCF data assimilation could nevertheless be used to correct for the instances where
the modelled fresh snow line is obviously wrong, as in May 2019 (Fig. 7).

The ellipsoidal patterns of the WSL/SL biases with respect to aspect (Fig. 8) demonstrate the potential of using WSL and SL retrievals in combination as a basis for a detailed model performance diagnosis. In most MEZ, SL biases in the eastern slopes are often substantially higher than in the western slopes by 50 to 200 m. The corresponding WSL biases, however, are in the
same order of magnitude. This finding suggests a deficiency in the model's representation of wind-driven snow redistribution processes. In our climate, dominated by west winds, these processes typically lead to an uneven distribution of snow between west and east exposed slopes, a phenomenon that the model currently fails to adequately represent. Such a finding may motivate better inclusion of wind redistribution processes in FSM (e.g. Vionnet et al., 2021; Baron et al., 2023; Quéno et al., 2023) even at the considered spatial resolution of 250m.


Conversely, the wet snow line bias in our simulations is still dominated by a South-North contrast. Given the strong influence of albedo on the wet snow line, we cannot exclude that further improvements are required with respect to the representation of snow albedo. Snow metamorphism, and solar angle indeed have strong effects on the albedo, which are not accounted for in the model (e.g. Gardner and Sharp, 2010), and result in topographic variability of the albedo as evidenced by satellite imagery
(e.g. Mary et al., 2013; Cluzet et al., 2020). The rather simplistic temperature-dependent time decay of albedo within FSM (Essery, 2015) cannot capture these processes in adequate detail, leaving potential for mitigation by way of data assimilation.

### 4.3 Complementarity with flat-field snow depth observations

As evidenced by Fig. 4, fsm_preop performs remarkably well against data measured at our 444 flat field snow monitoring sites.
There is little evidence and incentive that the model should be improved from this data alone. The remote sensing data reveals model shortcomings we would have remained unaware of. Wet snow maps directed us towards a possible solution of an issue that was also visible in our snow line comparisons, demonstrating the complementarity of all three datasets involved in our model performance assessment.

Additionally, flat field snow depth observations are scarce in Switzerland above 2500m a.s.l. (e.g. Cluzet et al., 2022). Likewise,
precipitation and air temperature measurements used to constrain meteorological input via assimilation or debiasing are scarce above these altitudes (Isotta et al., 2014; Vernay et al., 2022). Alpine areas are also where wind redistribution is prominent due an enhanced exposition to winds (Dujardin and Lehning, 2022), not to mention steep terrain and glaciated areas. All in all, for these high elevations, the uncertain input data and rare validation measurements is a challenge for achieving accurate snowpack simulations. Therefore, wet snow retrievals and snow cover fraction observations offer a unique opportunity to observe the





snowpack where other data is typically scarce, and provide key information to improve modelling and process understanding at high elevations. Nevertheless, a limitation arises from the challenges associated with distinguishing between snow-free and snow-covered glaciers in the snow cover fraction products.

    Finally, Fig. 4 shows that the significant improvements we obtain with fsm_optim in terms of WSL and SL come at the cost
of slightly deteriorating the model performance at flat field snow depth stations in the ablation period. This is not necessarily a surprise. fsm_preop inherits its tuning from model versions specifically calibrated against flat-field snow depth stations (Mott et al., 2023). The performance of fsm_preop against snow depth observations already establishes high standards. Additionally, it is a well-known fact that snow depth tends to be higher in flat terrain compared to the surrounding slopes (Grünewald and Lehning, 2015), a factor accounted for in our model through a precipitation multiplier (see Sec. 2.3). Further combining
satellite observations of wet snow and snow cover fraction with flat-field snow depth observations could help refining the relative behaviour of models in flat terrain with respect to the complex topography that predominates in mountain areas.

## 5   Conclusions

This work investigated the potential of wet snow retrievals from Sentinel-1 SAR to inform operational snowpack simulations in alpine areas over large extents. We could demonstrate that a pre-operational version of the snow-hydrological model fsm_preop
was well aligned with snow depth observations at hundreds of monitoring sites. However, notable discrepancies were observed between fsm_preop and wet snow maps at the onset of the melting season over a span of five years. For an in-depth performance assessment, we used the wet snow line (WSL) as an aspect-dependent elevation line across which the snowpack transitions from dry to wet snow conditions for a given hydrological unit. This metric provides a spatially aggregated summary of the information obtained from wet snow maps, enabling the exploration of regional variations in moistening dynamics. This
aggregation also mitigates data quality issues and data gaps at the pixel level that would hamper model evaluation.
While the simulated WSL matched the observations well in north slopes and flat areas, the model had a substantial dry bias in the south sector, suggesting a delay in the sun-induced melt. Snow lines (SL) derived from Sentinel-2 snow cover fraction retrieval show that this dry bias coincides with a too late seasonal snowpack depletion. Both biases were substantially reduced when an update of the model albedo parametrization was applied. This demonstrates that the wet snow maps contain comple-
mentary information for improving snowpack models.

    Wet snow maps provide valuable information about the snowpack state much earlier in the ablation season than snow cover fraction observations. This is an important asset of WSL data for operational applications. Snow cover fraction observations are much more detailed, but come too late in the season to allow mitigation of biases that arose much earlier during the winter.
Our study demonstrates that wet snow maps and snow cover fraction observations can provide complementary information to those from flat-field snow depth observations, allowing us to evaluate snow-hydrological models from a new perspective. Satellite observations can be used to evaluate the simulations in a diversity of topographic conditions, contrary to flat-field



snow depth observations. Finally, as snow depth observations are rare at high elevations, satellite imagery provides a unique opportunity to constrain models particularly at those altitudes.

*Code availability.* The GitHub repository of FSM2oshd will be made available in the published version of the manuscript.

*Author contributions.* BC and TJ conceptualized the study. BC performed the simulations and analysed the results with input from TJ. JM and LQ helped with running the simulations. All authors contributed to model development of the OSHD modelling framework and contributed to the manuscript.

*Competing interests.* The authors have no competing interests to declare.

*Acknowledgements.* The authors would like to thank Marie Dumont for helpful discussions on the albedo parametrization. This study has been partly funded by the ESA EXPRO+ AlpSnow - Alps Regional Initiative project (Contract No. 4000132770). The OSHD model (FSM2oshd) development and implementation was largely funded by the Swiss Federal Office for the Environment (FOEN).



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

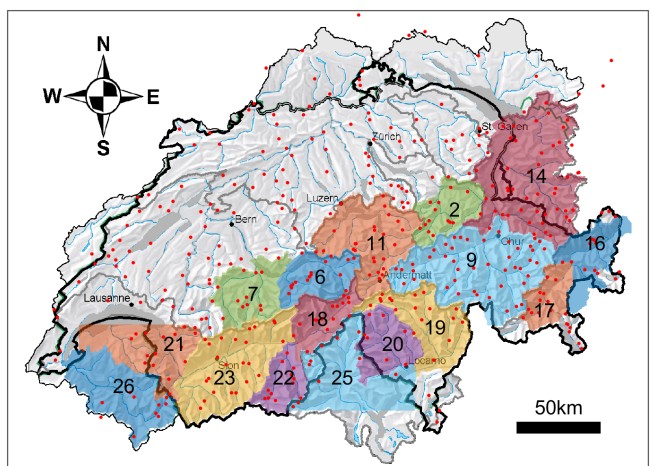

**Figure 1.** Map of the model domain, with hydrological units marked, and locations of the snow stations (red points). Switzerland's border is denoted by the thick black line.





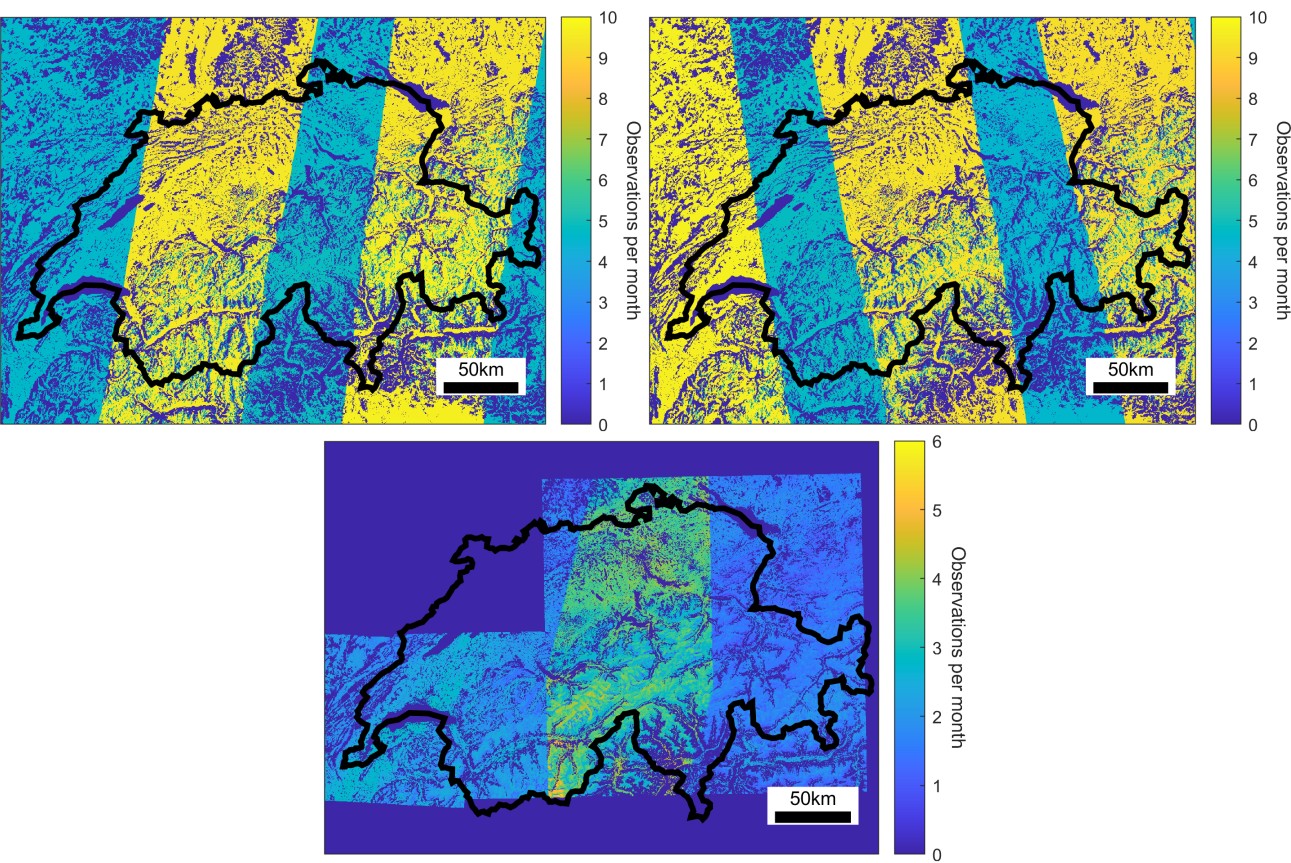

**Figure 2.** Sentinel-1 05UTC (top left) and 17UTC (top right), and Sentinel-2 (bottom), number of valid observations per month and per pixel over the whole period. Switzerland national borders are indicated in black.

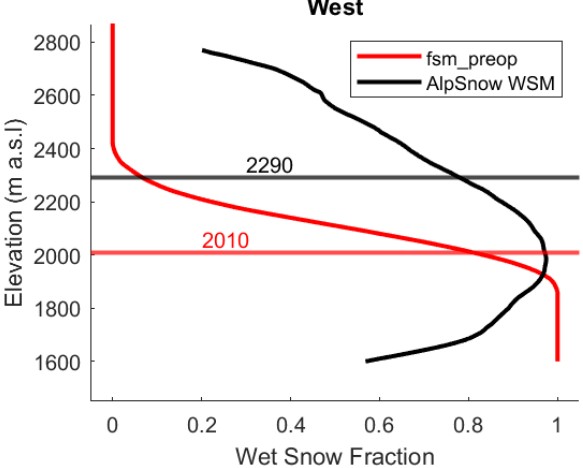

**Figure 3.** Computation of the wet snow line, in this example for westerly aspects of MEZ 11 on 2019-04-01, 17UTC.





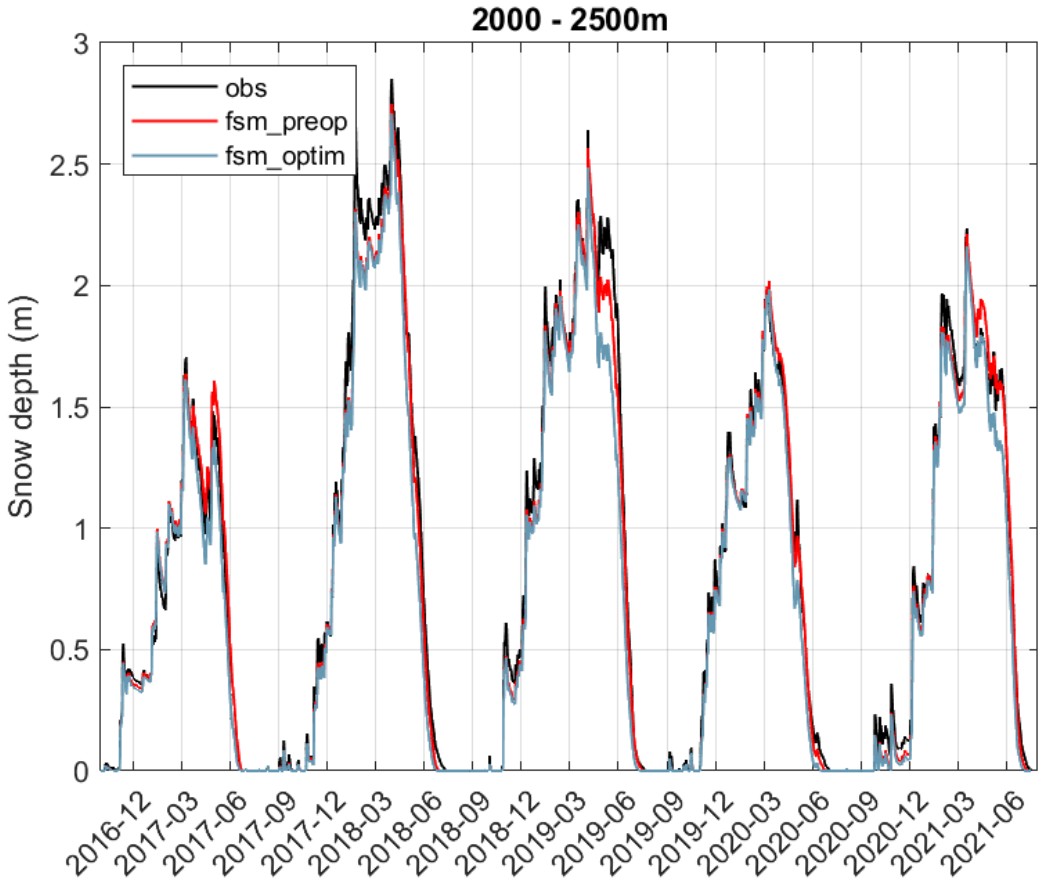

**Figure 4.** Average of fsm_preop (red), fsm_optim (light blue), and flat-field snow-depth observations (black) over the full study domain in the elevation range 2000–2500 m a.s.l.





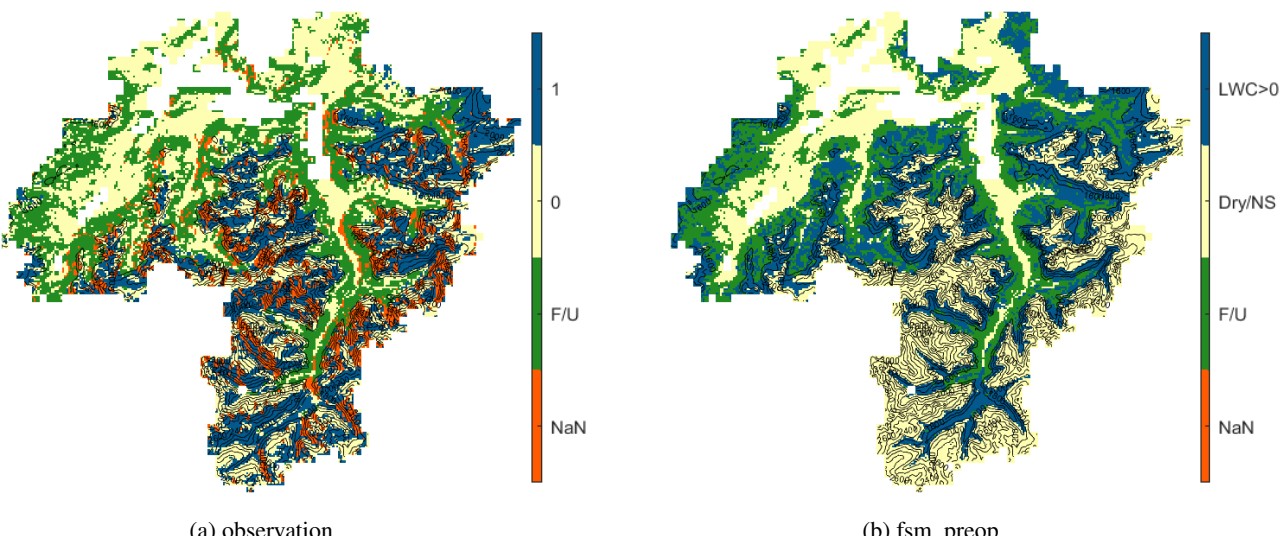

(a) observation  (b) fsm_preop

**Figure 5.** Sentinel-1 250m wet snow map (a) and simulated presence of liquid water using fsm_preop (b) for MEZ 11 on 2019-04-01, 17UTC. Blue denotes wet snow, light yellow dry conditions (or no snow/patchy snow), green forest and urban pixels, and orange denotes missing data due to geometric distortions in the observation. Lakes and pixels outside of MEZ 11 are masked out.

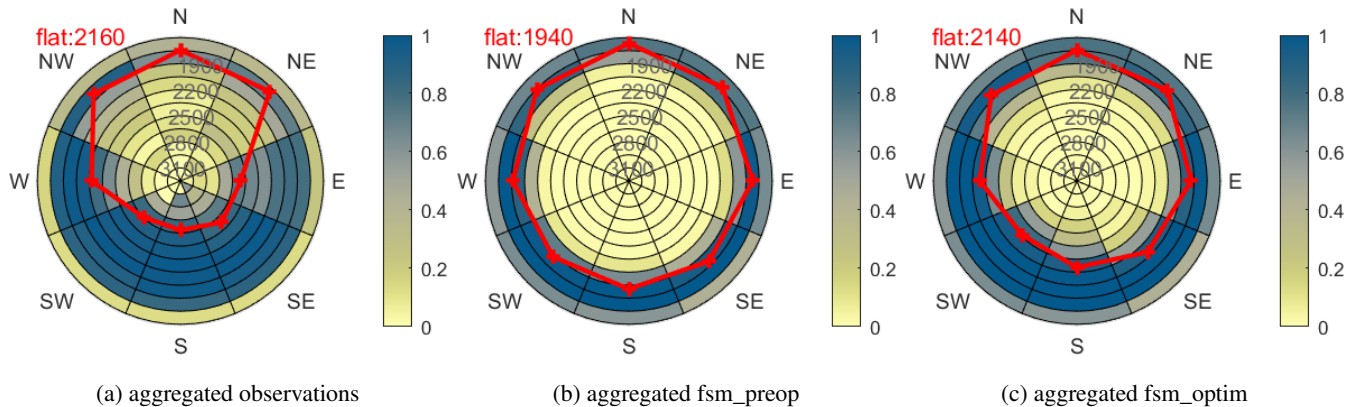

(a) aggregated observations  (b) aggregated fsm_preop  (c) aggregated fsm_optim

**Figure 6.** Fraction of wet snow pixels by 150m elevation bands and 8 aspect classes, from AlpSnow WSM (a) fsm_preop (b), and fsm_optim (c) over MEZ 11. The wet snow line is displayed in red.



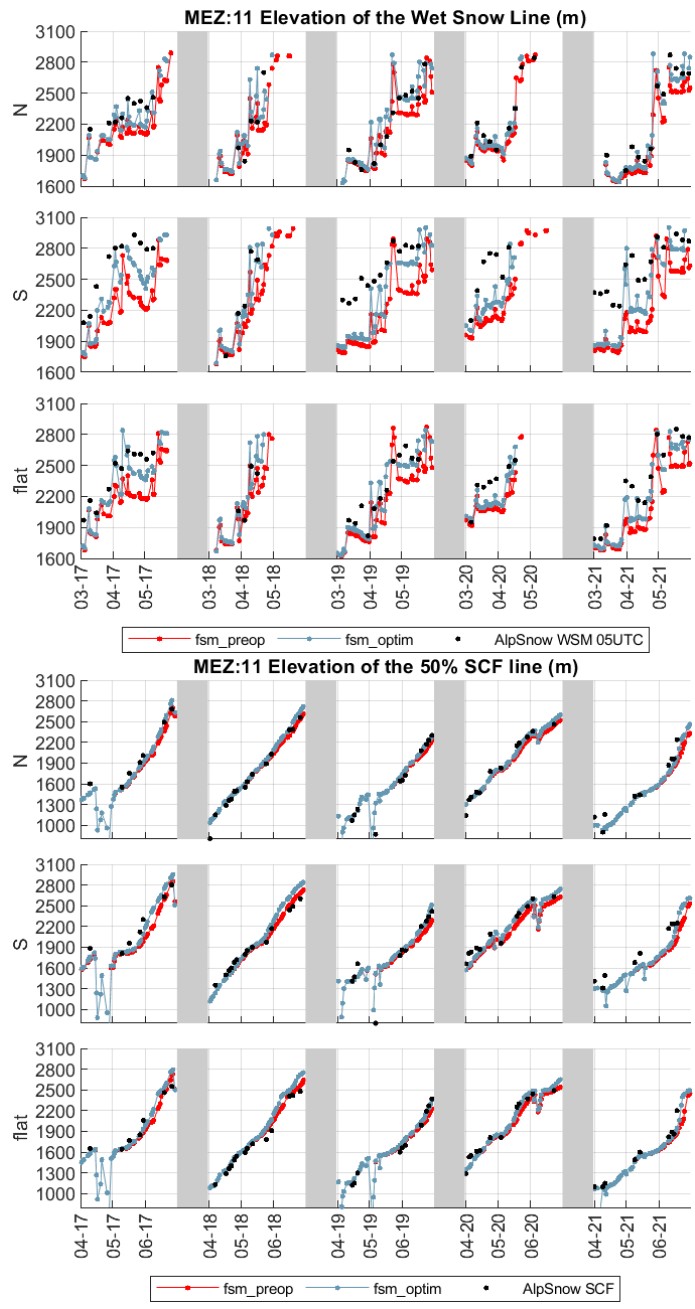

**Figure 7.** Time series of wet snow line (top panel) in the observations (black), fsm_preop (red) and fsm_optim (light blue) and snow lines (bottom panel) for the different aspects and seasons in MEZ 11. Only descending (05 UTC) Sentinel-1 tracks were used to determine the WSL.



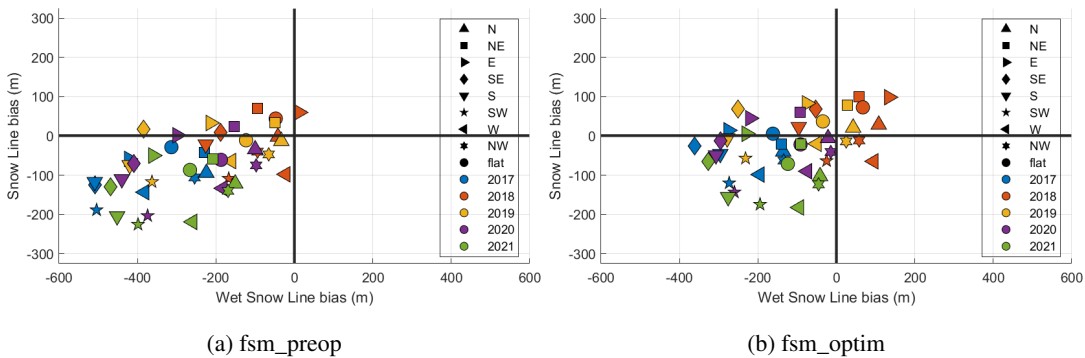

(a) fsm_preop                    (b) fsm_optim

**Figure 8.** Snow line bias as a function of wet snow line bias (at 05UTC) for MEZ 11, across years and aspects.

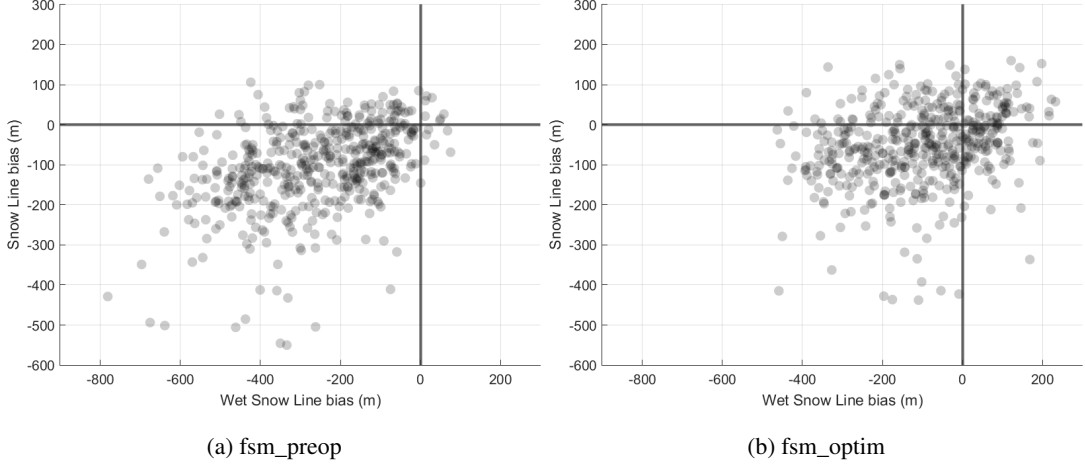

(a) fsm_preop                    (b) fsm_optim

**Figure 9.** Snow line bias as a function of wet snow line bias (at 05UTC) for all MEZ, across years and aspects.