# Peer review of "Using Sentinel-1 wet snow maps to inform fully-distributed physically-based snowpack models"

_EGUsphere, 2024_

## Author Comment (AC1)

Response letter for to first review round of:

**Using Sentinel-1 wet snow maps to inform fully-distributed physically-based snowpack models**

Bertrand Cluzet[1], Jan Magnusson[1], Louis Quéno[1], Giulia Mazzotti[2,1], Rebecca Mott[1], and Tobias Jonas[1]

[1]WSL Institute for Snow and Avalanche Research SLF, Davos, Switzerland
[2]Univ. Grenoble Alpes, Université de Toulouse, Météo-France, CNRS, CNRM, Centre d'Études de la Neige, Grenoble, France

In the following, comments from the reviewer appear in black. The authors' answer is in blue, with planned changes in italics.

Reviewer #1

This study explores using Sentinel-1 data to enhance the accuracy of fully distributed, physically-based simulations of mountain snowpack. Specifically, the wet snow line (WSL) altitude derived from Sentinel-1 wet snow maps and the snow line (SL) altitude derived from Sentinel-2 snow cover fraction maps were compared against the WSL and SL simulated by the FSM2oshd model.

The study demonstrates good agreement between the model and ground observations, primarily located on flat terrain, in terms of snow depth. However, significant discrepancies emerge between the modeled WSL on south-facing slopes and the WSL derived from Sentinel-1 data. This discrepancy appears to correlate with errors in the model SL detection. By adjusting the fresh snow albedo parameter, the authors achieved an improvement in model performance for south-facing slopes when compared to Sentinel-1 WSL. This highlights the value of satellite data in identifying and correcting model biases.

The core concept of leveraging satellite data for model evaluation and improvement is highly promising and opens doors for advanced data assimilation techniques. However, further research and elaboration on the manuscript are needed to address some limitations and strengthen the work before publication.

Thank you for your positive feedback on our study and your valuable comments for improving the manuscript. Below, we have provided our responses to your comments and outlined the changes we plan to implement to enhance the paper.

A critical issue I see with the paper is how the authors arrived at the conclusion of adjusting the fresh snow albedo parameter as the primary explanation for the underestimating melt on south-facing slopes. Indeed, the explanation provided in the manuscript lacks a logical progression (i.e., the information is sparse around the paper) and fails to justify the selection of fresh snow albedo value as the sole "culprit".

If fresh snow albedo, or in general albedo, is a significant factor, the work should have explored incorporating satellite-derived albedo values. In fact, research on satellite-based albedo estimation for snowpack data assimilation is an already established field. Reviewing and citing relevant past studies would strengthen the paper and provide valuable context, which are now omitted. Moreover, by comparing the findings of this study with previous work on assimilation of satellite-derived albedo, the authors could highlight the advantages and disadvantages of each approach. This analysis would allow readers to understand the potential benefits of using Sentinel-1 data for assimilation compared to traditional albedo observations. However, this work offers an opportunity to explore alternative approaches beyond this traditional reasoning.

We agree that remotely sensed albedo products could be used to test this hypothesis. However, to our knowledge, the accuracy of state-of-the-art albedo retrievals in complex terrain is not sufficient. In the mountains, such products typically suffer from aspect-dependent biases (Cluzet et al., 2020), and their accuracy does not reach requirements (typically less than 5% errors) to inform state-of-the-art snowpack models (Revuelto et al., 2020). Albedo retrieval in complex terrain is a challenging inverse problem, which requires a very accurate estimation of the incoming radiation to the pixel, including indirect illumination from neighboring pixels (whose surface properties introduce a coupling term in the inversion) as well as atmosphere radiative properties (Lamare et al., 2020). Therefore, biased albedo observations constitute a considerable challenge for data assimilation (Cluzet et al., 2020), as evidenced by the limited number of publications around that topic in the review from Largeron et al., (2020). In summary, we consider that satellite derived albedo products do not qualify yet for detailed model evaluation and data assimilation in complex terrain.

Please note further, that this work is ultimately about the information content and utility of remotely wet snow maps in the context of physically based snow modelling. Fresh snow albedo was just identified as a suitable tuning parameter in our example of demonstrating how the concrete nature of information available from wet snow data could lead to mitigate a model shortcoming that we had so far been unaware of. However, this work never attempted to be an exhaustive model evaluation exercise, nor on the assimilation of satellite-derived albedo data.

The perception by reading the manuscript is that the suggestion to work on the fresh snow albedo values seems like a convenient shortcut, addressing numerous potential sources of errors in the snowmelt model with a single parameter tweak, without the complexity to perform a data assimilation inside a physically based model. It is interesting to review some of the sources of errors. These include, for example i) shortwave radiation spatialization; ii) albedo decay function; iii) longwave radiation calculations and spatialization; iv) snow temperature; v) water transfer processes; vi) distinguishing liquid from solid precipitation, etc. While the paper mentions some of these sources of errors sparsely around the different part of the manuscript, a dedicated section with a clear, step-by-step reasoning process is crucial for readers comprehension.

We agree that the discussion could be more condensed in one place. Analysis of wet snow maps have revealed a very specific aspect and slope dependent model bias pattern. This choice may only seem convenient, but it ticks many boxes unlike, e.g., iii), v), vi), which have no practical utility in fixing aspect dependent model biases (more details at the end of the comment). On the other hand, in the absence of complementary and much more detailed datasets, we cannot identify whether the model biases originate

from shortcomings in the fresh snow albedo, the albedo decay function, or both. We decided in favor of the former solution, but do not exclude the possibility that a similar performance improvement could be found by adjusting the albedo decay rates instead. In the revised manuscript we will improve the justification of our choice and extend the discussion accordingly.

The following block will be added to Sec. 2.3 (hence improving the description of the albedo parameterization):
* * *
*FSM2oshd uses a re-calibrated version of the albedo parameterization from Douville et al., (2015). In the parameterization, the snowpack albedo decays linearly with time when the snow surface temperature is below zero °C and decays exponentially when the surface is melting. The minimal albedo value is 0.5. The constant decay times scales are $\tau_{melt} = 130h$ for melting snowpack surface, and $\tau_{cold} = 3000h$ for subfreezing surface temperature, reflecting the fact that wet snow metamorphism is quicker than dry snow metamorphism. In the presence of snowfall, the albedo is progressively refreshed to fresh snow values.*

*In FSM2oshd, the fresh snow albedo $\alpha_{fs}$ varies as a function of altitude z as follows:*

$$\alpha_{fs} = \begin{cases} 0.8 & if\ z \leq 1500 \\ \alpha_{fs}^{max} + \dfrac{2300 - z}{2300 - 1500}(0.8 - \alpha_{fs}^{max}) & if\ 1500 < z \leq 2300 \\ \alpha_{fs}^{max} & if\ z > 2300 \end{cases}$$

[Figure]

*Figure 1: Fresh snow albedo parameterization as a function of elevation for fsm_preop (red) and fsm_optim (light blue).*

*where the snowpack albedo α increases with hourly snowfall Sf (mm):*

$$\alpha = \begin{cases} \alpha_{fs} & if\ Sf24h > Sfmin \\ \alpha + (\alpha_{fs} - \alpha)\dfrac{Sf}{Sf_{min}} & otherwise \end{cases}$$

*This rule includes a threshold enforcing an albedo of $\alpha_{fs}$ for 24h-snowfall above Sfmin=10mm (Douville et al., 1995).*

*The albedo parameterizations in fsm_preop were tuned through an assessment against snow depletion rates in springtime measured at flat field sites (Mott et al., 2023), where $\alpha_{fs}^{max}$ was set to 0.92. A preliminary comparison with Sentinel-1 wet snow retrievals (c.f. Sec. 3.2), however, showed that in*

*fsm_preop simulation melting was initiated too late, with a particular focus in the south-exposed slopes. As we will show, this model shortcoming could be mitigated by lowering $\alpha_{fs}^{max}$ from 0.92 in fsm_preop to 0.86 in the so-called fsm_optim. Below, we will compare both model versions.*
* * *
The following block will replace the current discussion in Sec. 4.2 (hence focusing the discussion into one chapter):
* * *
*The wet snow line analysis evidenced a systematic dry bias in the pre-operational version of our model, fsm_preop (Fig 9a). The comparison of observed versus simulated WSL revealed a systematic pattern with model biases being particularly pronounced in the south exposed slopes. This pattern indicated the need for a model adaptation that would accelerate the initiation of snowmelt particularly in south-facing terrain, suggesting a mitigation strategy with focus on shortwave radiation. Since FSM is driven with shortwave radiation per inclined surface area (Mott et al, 2023), any reduction of the snow albedo will preferentially augment net shortwave radiation in south exposed slopes. Indeed, the adjusted albedo parameterization in fsm_optim, while being closer to literature values (Gardner and Sharp, 2010) resulted in improved simulations of wet snow and snow cover fraction (Figs 9b and 10b) without incurring any considerable impairment of the excellent match between observed and simulated snow depth at the stations (Fig 5). However, due to the lack of reliable spatialized albedo data in complex terrain (Largeron et al., 2020; Lamare et al., 2020; Cluzet et al., 2020), we can only speculate that this change actually addresses existing model deficiencies in the most efficient manner.*
* * *
In response to the reviewer's six suggestions i) to vi) regarding alternative mitigation strategies we note the following:

(i)      Shortwave radiation spatialization: There is no reason to believe that COSMO has an aspect-specific bias. HPEval (Jonas et al., 2020) used for further downscaling might introduce some topographic biases by neglecting reillumination from the opposing slopes. Including this effect, however, would preferentially increase net shortwave on the north facing slopes, which would counteract the observed bias pattern.

(ii)      Albedo decay function: Yes, reducing albedo can be realized by either reducing the fresh snow albedo, or the albedo decay rate. While here we deployed the first option, we acknowledge that similar results might be attainable using the second option. Accelerating the albedo decay instead caused at times some unrealistically low albedo values whilst still cold and dry, hence our preference for the first option.

(iii)      longwave radiation calculations and spatialization: very unlikely to cause the bias patterns and magnitude we observe.

(iv)      snow temperature: very unlikely to cause the bias patterns and magnitude we observe.

(v)      water transfer processes: very unlikely to cause the bias patterns and magnitude we observe.

(vi)      distinguishing liquid from solid precipitation: very unlikely to cause the bias patterns and magnitude we observe, further a systematic bias in the phase of precipitation would certainly render the good model performance at the stations impossible.

In fact, a more thorough exploration of these concepts could likely lead to the conclusion that using WSL data can be potentially used in a more effective way than solely operate on the fresh snow value. This seems to be also confirmed by the results: Figures 8 and 9 suggest that the change in fresh snow albedo only partially improves the results and there is still room for improvements. For this I suggest the authors to explore the different methods to assimilate WSL data into the model. This can be done operating on other model parameters, as done for fresh snow albedo, or even on the physical equations. For example (just to stimulate the discussion), one alternative to modifying albedo is introducing a new parameter linked to the Sentinel-1 observed WSL that control the net radiation. This parameter would be meant to adjust the net energy input for some of the different sources of error mentioned before offering several advantages. It can indirectly account for various model errors without directly modifying a specific physical parameter like albedo. This avoids potentially unrealistic changes to an established physical value. By focusing on the observed WSL, this approach may be more effective than relying on temporally sparse albedo observations coming from optical imagery or changing the fresh snow parameter. It directly adjusts the energy budget to reflect the presence of wet snow for a specific aspect and altitude, which Sentinel-1 data can reliably identify. This could lead (or not, this is just an example to stimulate the discussion) to a more robust simulation of snowpack melt showing that optimizing WSL can improve SL and snow depth, which is not an immediate logical consequence (see next paragraph). I would suggest the Authors expanding the discussion on these aspects in the manuscript. This will provide a clearer message for the (remote sensing) community: not only albedo, snow cover area or snow depth are important and feasible to be assimilated.

As mentioned in response to your previous point, there are good reasons to believe that mitigating the observed model bias patterns by adapting the albedo parameterization is our best option. Moreover, we lack complementary and much more detailed datasets that would be needed to gauge alternative solutions. To our best knowledge, unbiased albedo observations at high spatial and temporal resolution don't exist for Switzerland. As you suggest, assimilating wet snow lines could be an option. However, without additional data, WSL observations alone do not provide information about when, where, and why respective model biases have been caused. In this situation, data assimilation of WSL would only mitigate remaining biases between years (i.e., year specific vs constant albedo tuning, c.f. Figure 9), but we would be unable to draw any further conclusions beyond our current solution using albedo without running into new equifinality issues.

Changes to Sec. 4.2 make it clear why we currently do not consider remote sensed albedo products. The updated section 4.3 is also more explicit in explaining the current limitations and potential of the wet snow line observations (and wet snow maps in general) for the purpose of data assimilation and model evaluation.

In this sense, the insightful analysis of the connection between WSL bias and SL bias, presented in the paper, could benefit from a more thorough integration into the discussion above. In general, WSL primarily reflects the cumulative energy input received by the snowpack over time, while SL integrates various snowpack processes throughout the season, primarily including snowfall, redistribution, melting and sublimation (and evaporation). By assuming the precipitation and snow redistribution are accurately represented in the model, WSL information may help refine the energy distribution within the snowpack.

This could occur when factors like snow temperature, albedo, slope, aspect, or rain-on-snow events are not adequately accounted in the spatialization operations. By correcting using the observed WSL, the energy input is adjusted, potentially leading to a more accurate simulation of the overall snowpack depletion and, consequently, a better representation of the SL throughout the season. However, it is not immediately clear how solely adjusting WSL leads to improved SL, and therefore improved snow depth (or SWE) across the entire season. I would suggest the Authors expanding the discussion on these aspects (some of them are already in the paper). This would significantly strengthen the paper.

*Thanks for the suggestion. We have extended/focused the discussion by entering the following paragraph to Sec 4.2:*

*Simulated WSL and SL represent the equivalent elevation of two timing variables, i.e. the first occurrence of wet snow and the melt out date. Both are a consequence of the coupled energy and mass balance equations and are affected by changes in the snow albedo in the same way: a reduction of the albedo will prepone the timing of both events, entailing a mechanism why WSL and SL biases are correlated. Obviously, there are other factors that contribute to model biases of each, so that such a correlation is not perfect. Apart from the energy turnover, there is also a direct link between the occurrence of wet snow and the progression of snowmelt. The occurrence of wet snow accelerates the albedo decay through wet metamorphism (see Sec. 2.3), which in turn increases the net shortwave input to the snowpack.*

*Observations indicate that both WSL and SL simulated by fsm_preop were typically biased low. Being able to improve both by decreasing the albedo is therefore expected. Remaining SL biases (Fig. 9) call for additional model improvements, requiring complementary data to the wet snow maps exploited in this study, spatially distributed and temporally resolved data of snow depth or SWE being the most obvious choice.*

In summary here are some areas for potential development:

Provide a more nuanced explanation of how WSL information may refine the melt within the model considering all the sources of errors.

*Addressed by the above answers.*

Elaborate on a conceptual strategy to potentially assimilate WSL showing the specificity of the assimilation (e.g., how would the assimilation parameter enter in the snowpack equation/parameter and how can be calculated based on the WSL observations?)

*Addressed by the above answers.*

Elaborate on the specific modeling conditions where WSL correction impacts SL across the season. Are the same conditions that allows WSL correction to improve snow depth simulations?

*Addressed by the above answers.*

Discuss potential limitations of assimilating WSL, particularly in scenarios where precipitation or redistribution might not be accurately captured.

Addressed by the above answers.

In conclusion, my suggestion is that by elaborating on these concepts throughout the manuscript would strengthen the paper and finally present a clear direction for future research focused on developing robust and innovative approaches to data assimilation. These new approaches should move beyond relying solely on "traditional" input variables like snow cover, albedo or snow depth. To reflect this broader focus, the authors can consider changing the title of the paper to highlight its exploratory nature e.g., Exploring how Sentinel-1 wet snow maps can inform fully-distributed physically based snowpacks models.

We adopted your title suggestion.

Title changed to:

*Exploring how Sentinel-1 wet snow maps can inform fully-distributed physically based snowpacks models.*

Detailed comments:

L40-41: There are a few points to consider regarding the terminology used to describe melting processes. The term "moistening" has a specific definition introduced by Marin et al. (2020), building upon earlier works by Dingman (2015) (where the melting phases classification use a slightly different taxonomy) and Techel & Pielmeier (2011). Using this term without proper context or referencing the original work might be unclear for readers unfamiliar with that definition.

We added the suggested references.

When referring to wet snowpack, "saturation" typically indicates the percentage of the porosity filled with liquid water. However, in this context, it seems that the authors meant the maximum capacity of holding water of the snowpack.

This sentence was edited to include this nuance:

*Once the amount of liquid water in the upper layers of the snowpack reach the maximum holding capacity,*
*…*

Finally, the process of snow ripening is slightly more complex than a simple "bucket scheme" (e.g., Marin et al. (2020), Techel & Pielmeier (2011), Essery (2015)). Here is a more accurate description: the liquid water released or absorbed from the superficial layers gets in contact with the subfreezing snow present

underneath and freezes. This releases latent heat that causes the snowpack to warm up. This starts the process of snow ripening.

We extended the description as requested:

*…holding capacity, the liquid water released or absorbed from the superficial layers gets in contact with the subfreezing snow present underneath and freezes. This releases latent heat that causes the snowpack to warm up. This starts the process of snow ripening.*

My suggestion is that when dealing with specific terms or concepts, it is always good practice to cite the original (and most recent) papers. This allows readers to delve deeper into the topic if they wish.

L66. Margulis et al 2016 is only one example.

We added a reference to Fiddes et al., (2019) and added the nuance "through data assimilation", since purely observation based products are not considered here:

*This information can be used to reconstruct snow water equivalent through data assimilation (e.g., Margulis et al., (2016), Fiddes et al., 2019).*

L67: why the information is qualitative and not quantitative? Maybe binary?

We argue that this information is qualitative, and not quantitative, in the sense that it cannot be trivially related to liquid water content. Indeed, the backscatter drop threshold is somewhat arbitrary, and the resulting classification is affected by a multitude of factors, from the remote sensing and signal processing, surface roughness property, to the detailed snowpack stratigraphy of liquid water content and physical properties, which are not fully understood yet as you detail in your paper (Marin et al., 2020). This nuance is important to us, because we think that it would be misleading to treat this information as qualitative and assimilate it directly as a measure of "liquid water content". This terminology is consistent with our approach which tries to correct obvious mismatches with the model (by using a zero threshold on the model's end), while not pretending to go into the details of the signal itself.

L70: Premier et al 2023 used S1 information for assimilation. However, this was done in reconstruction.

This sentence was changed into:

*Key to operational snow-hydrological model applications, wet snow retrievals can anticipate the onset of snowmelt, making them a potentially interesting data source for assimilation (Malnes et al., 2015) or observation-based reconstruction (Premier et al., 2023).*

L90: While explicitly stating research questions is a good way to introduce a paper, it requires more specific details in this case. For example, snow depths, snow cover fraction and wet snow status are not interchangeable concepts (even if connected), how do you check if they are complementary? Moreover,

to test if the wet snow information enhances the accuracy of the snowpack simulations imply to use independent validation data in complex terrain, which seems not to be the case.

The research questions were deleted as the aim of the study is already clearly stated at the beginning of the paragraph and reformulating questions as well as explicitly answering them would complicate and lengthen the paper.

Finally, I suggest the importance of answering the research questions explicitly in the conclusion. The conclusion should summarize the key findings related to each question, demonstrating whether and how Sentinel-1 data improves the model representation of snowpack.

See answer above: the questions were removed from the revised manuscript.

L103: It would be important to explicitly mention that all these stations are in flat terrain, right?

We added "measured on a flat surface" picking this formulation to stress that the terrain is locally flat, but not necessarily the corresponding 250m pixel.

*There are 444 locations with daily in-situ snow depth observations measured on a flat surface available throughout the domain indicated by the red dots in Figure 1.*

I personally would like to see also the distribution by elevation and predominant aspect in the final resolution of 250m.

Here, we include a histogram of elevation and aspect (for slope > 8 degrees) over the 16 considered hydrological units (Fig. 1). They show that no aspect is strongly under/overrepresented, and that the elected elevation band (1600-3200) corresponds to the bulk of the elevation histogram. We don't think that it's necessary to include this in the manuscript.

[Figure]

*Figure 2: histogram of aspect (left panel) and elevation (right panel) for the model grid cells outside of glaciers and with less than 80% forest cover.*

L113: not only patchiness but also superficial snow roughness is playing a role.

Thanks for pointing that out. The sentence was slightly reformulated:

*As evidenced by Marin et al., (2020), the backscatter signal resumes to values acquired under dry snow conditions as the snowpack becomes rough and patchy throughout the ablation season, even though at that time the snowpack is usually fully saturated with liquid water.*

L115: For me it is not clear what is meant with "fully saturated". Maybe the Authors refer to the fact that the snowpack exceed the maximum water holding capacity? If so, this is not always the case. If there are no impermeable barriers, LWC is typically limited by the snow grain density and shape (Goto et al., 2012). It may be that with large grains the snow can have larger porosity. In this case the LWC can be relatively low, since LWC is drained quickly by gravity. Instead, if the Authors with "fully saturated" means 100% water saturation, means the snowpack is (decomposing) snow slush. And this is not the case for patchy snow. Please clarify this.

By "fully saturated", we mean that the entire ("full") column is at the local maximal liquid water holding capacity ("saturated"), which in essence, as you say, depends on local snow microstructural properties. Our intention here is mostly to re-explain (following your 2020 paper) the paradox of a relaxing backscatter in the ablation period although the snowpack is substantially wet. We need to stress that the total amount of liquid water content remains substantial in the snowpack column throughout the ablation season, which may seem counterintuitive to some readers.

We reformulated into:

*"The snowpack column is usually entirely saturated with liquid water."*

L116: "viewing slant angle": due to the SAR lateral view.

Reformulated as requested:

*Furthermore, due to the slant angle under which SAR acquisition is performed, …*

L116: masking the geometric distorted regions cause data gaps. Layover and foreshorting does not mean the data is missing (like in shadow), the backscattering of these area is "distorted" inside the selected final resolution cell.

We agree that our formulation was not accurate enough. We reformulated into the following. The use of 'result' implicitly applies to literally missing data (shadows) or required masking (geometric distortions).

*Furthermore, due to the slant angle under which SAR acquisition is performed, geometric distortions (layover and foreshortening) as well as shadowing result in data gaps in rugged terrain.*

L118: the nominal revisit time at equator is 6 days for the two satellites. Sentinel-1B failed in December 2021. Due to track overlap, at the considered latitude, a minimum of 2 tracks (one ascending and one descending) to a maximum of 4 tracks (2 ascending and 2 descending) are available depending on the ground location, for every repetition cycle of 6 days when the two satellites were available, and every 12 days when only Sentinel-1A is available. A reference to Fig 2 can be used here to better explain this concept.

*We added a reference to Fig. 2. Note that in the dataset that we have, some orbits are not available (we assume that these correspond to higher zenithal angles for which geometric distortion and shadowing are larger) and therefore in most of the domain we only have one ascending and one descending track.*

L118: are the maps binary? Wet, non-wet?

*The binary nature of this product was mentioned on l. 113, however we added this information here as well. Changed to:*

*AlpSnow binary wet snow maps, …*

L121: with "coinciding" is it meant "aligned to the upper left cell of the model grid"?

*We reformulated into: "Wet snow maps were aggregated into pixels of 250m corresponding to the cells of the model grid.*

L123: Had the forest areas been masked out from the AlpSnow products before use?

*Yes, but we additionally applied a more detailed forest mask that was also used for our FSM2oshd model simulations.*

*To clarify, we added the following sentence:*

*These products were additionally masked using a 10m forest mask (Mott et al., 2023) which is more detailed than the forest mask already applied to the raw data.*

L125: it would be good to express the % in number of pixels. In the end you aggregated inside a window 5x5 pixels, right?

*Correct, this is indeed the case. We changed the sentence as follows:*

*For each 250m pixel, an aggregated binary value was computed from the twenty five 50m subpixels as follows:*

L128: Do you expect S1 provides a good sampling time for wet snow?

Morning and evening overpasses offer different and interesting perspectives on snowpack moistening. As conceptually suggested by Marin et al., (2020), the evening overpass will react to diurnal superficial moistening, while the morning overpass will give a wet signal only once the melt is strong or deep enough that it cannot entirely refreeze overnight. For this reason, we mostly focused on morning overpasses in this study.

L135: following problems? I suggest expanding the listed items with a clearer description of each point.

Reformulated into:

*Based on optical data, these products are inherently affected by the following problems (Barrou-Dumont et al., 2021; Keuris et al., 2023)*

- *it is difficult to discriminate clouds from snow, and to delineate cloud shadows.*
- *undetected thin cirrus clouds may cause a decrease of the retrieved snow cover fraction.*
- *shaded snow in steep terrain and sparse forests, may also be affected by negative biases in SCF, in particular from November to March when shadow extents are larger (and there is snow on the ground)*
- *nondiscrimination of snow and bare ice, causing snow-free glaciers to be classified as snow.*

L139: what is a snow-free glacier? Bare ice is identified as snow? Please clarify.

This was clarified (see previous point)

L151: I suggest recalling some basic details of the model in this section e.g. how long- and short-wave radiation is accounted in the net radiation budget?

Snowpack energy budget is computed as usual for physically-based snowpack models. Downwelling longwave is a model input. Upwelling longwave is determined from the surface temperature, which is iteratively computed to close the energy budget. Shortwave input is provided per inclined surface. The shortwave radiative budget is computed from the snowpack albedo. This should be clearer now that significant elements have been added to the description of the shortwave radiation part in Sec. 2.3.

L170: The rationale behind adjusting fresh snow albedo from 0.92 to 0.86, and how these values compare to the literature, needs further clarification.

The added elements in Sec. 4.2 of the discussion, along with a thorough description of the albedo parameterization and the impact of an updated parameterization (see pages 2-4 of the response letter) should answer this comment.

L172: I suggest introducing a conceptual block scheme of the operation done in the comparison.

We think that the comparison we perform is conceptually rather simple and exemplified by Fig. 3, 5 and 6 (as per the old figure numbers) so we don't think it's necessary to lengthen the paper with such a figure.

L176: I generally agree that the Sentinel-1 is very sensitive to small LWC, but this need to be quantified. What is the value you used?

This threshold value is important indeed, and we thought that the formulation we used, "as soon as liquid water was present in any of the snow layers" was clear enough, but we reformulate this:

*Given the high sensitivity of the observation to small liquid water amounts in the snowpack, we considered modelled snow to be wet as soon as the total liquid water content was strictly above zero…*

L180: While this is true, it is important to acknowledge that aspect and slope can also influence the melting as shown later.

This is what we meant by "which primarily depends on topographic factors", but we can be more explicit:

*… which also depends on topographic factors such as aspect and slope.*

L190: I missed where are the glacierized areas.

Forest and glacier masks were added to Fig. 2 for illustration.

L191: 300m altitude? Please better clarify also considering the previous aggregation at 250m and how the two aggregations may interact.

Yes, this is an elevation window, we added a clarification:

*…using a window width of 300m of elevation…*

The two aggregations do not "interact", the purpose here is just to compute an elevation-dependent proportion of wet snow pixels for a given aspect, from which we derive the wet snow line.

L196: how the value 0.8 times WSFmax was selected?

This value has been determined manually in order to accommodate the specificities of the observed wet snow fraction (see Fig. 3) as a function of elevation:

1. The maximum observed wet snow fraction does not necessarily reach 1 (hence the 0.6 threshold l. 194 to determine whether a snow line is to be searched or not). For this reason, the threshold must be relative to the maximal value.
2. The raw AlpSnow retrieval exhibits false positives on ridges which cause a "plateau" of relatively high wet snow fraction at high elevations. Therefore, our threshold must be above 0.5 x the maximal wet snow fraction.

We will specify that the threshold was determined manually:

*…0.8 x WSF_max. This value was optimized manually. If it was not reached…*

L224: how to see flat terrain from Fig 6?

The altitude of the wet snow line in flat terrain is written in red in the top left of each panel, we clarified this in the caption:

*…The wet snow line is displayed in red, and its altitude in flat terrain is indicated in the top left of each panel.*

L226-230: This explanation cannot be introduced in the results sections, but before in the manuscript. While adjusting fresh snow albedo can be a way, it is worth considering whether directly modifying net radiation might be a more comprehensive strategy as indicated by this sentence.

The edited Section 2.3 does address this remark.

L270: Again, operating on the fresh snow albedo value does not seems the correct indication to give here.

The edited Section 2.3 does address this remark.

L294: While I can generally agree with these statements it is important to acknowledge the limitations of spatial aggregating and reduce the dimensionality using the snow line concept. These operations inherent overlook some important spatial variations. For example, areas with sub-zero air temperatures and specific topographical features, like the shadowed bottom of a narrow valley, might remain dry even when higher altitude snow covered areas may be wet. Additionally, it should be acknowledged that by exploring high-resolution data or alternative spatial modeling techniques might be valuable for capturing these finer-scale variations.

Indeed, in essence, topographic aggregation makes the signal more robust at the cost of an information loss: this is a compromise, that we now mention more explicitly in the first paragraph of Sec 4.1 (l. 288-290 of the initial submission version):

*At the same time, while some relevant information from individual pixels may be lost, key spatial information was maintained by resolving biases in terms of region, elevation, aspect, and slope, which has been shown to be of critical relevance (Cluzet et al., 2020).*

L295: especially for real time assimilation, it seems.

There might be a misunderstanding here, potentially in relation with a typo in the original sentence. The limitation applies to the binary nature of the wet snow maps products, which makes them impractical for

data assimilation in general, because of its nonlinear nature and the difficulty to specify observation errors ingestible by data assimilation algorithms in such a case. The sentence was edited into:

*…and modelling, while the direct assimilation of binary products such as wet snow maps (or snow masks) into physically based snow models is typically seen as unfavorable (Baba et al., 2018; De Lannoy et al., 2012)*

L316: please provide a reference.

We added a reference to Marin et al., (2020) (i.e., Sec. 4.2 of that paper).

L332: Premier et al 2023 as well. Using melting phases derived from Sentinel-1.

This is indeed an interesting reference that was missing (see addition above), but we would not add it here, as it is not strictly speaking an assimilation approach.

L346L maybe the word cluster fits better?

No, here we really refer to the ellipse that is drawn by the biases for a given years across different aspects, (l. 260-267). We will stress this by adding a reference to the corresponding section.

L387: for the conclusion I suggest answering the questions presented in the introduction. Do you plan to change the fresh snow albedo value in FSM2oshd?

The fresh snow value was indeed updated within our operational instance of FSM2oshd.

Fig. 1 In the caption and legend the "numbers" explanation is missing. Are them the MEZ IDs? Why only some are reported?

MEZs and their numbers are defined by the federal office of environment. Some of them are not represented because they are outside of the alpine area. For the sake of clarity, the caption was adapted to:

*Map of the model domain, with the identifier of hydrological units (MEZ), and locations of the snow stations (red points).*

Fig.2 It is not clear why some areas have 0 observations. I imagine shadow, layover, clouds? Please explain it in the caption.

These areas correspond to dense forest and water bodies, for the vast majority. The figure was edited to show forest and glacier masks, and the caption was edited:

*Sentinel-1 05UTC (top left) and 17UTC (top right), and Sentinel-2 (bottom), number of valid observations per month and per pixel over the whole period. Switzerland national borders are indicated in black.*

Fig. 3 What is the range of "westerly" aspect? Are the real curves as derived from the data or a conceptual example? What was the snowline for this case? Why S1 WSL is decreasing for lower altitude? Patchiness and snow roughness? Please explain it in the caption.

Thanks for the comment. Towards low elevations, AlpSnow observed wet snow fraction is decreasing as explained in the product description. The legend was edited accordingly:

*Computation of the wet snow line, for fsm_preop and AlpSnow data, in this example for westerly aspects (247.5-292.5degN) of MEZ 11 on 2019-04-01, 17UTC.*

And a sentence was added at the end of the second paragraph of Sec. 2.5, l. 191:

*Towards low elevations, the wet snow fraction from AlpSnow is decreasing probably because of snow roughness and a decreasing snow cover fraction.*

Fig 4. The model was calibrated using the 444 stations for the reported years or others years?

Yes, the model was calibrated using the 444 stations over winter years 2022 and 2021.

Fig.5 why fsm_optim example is not reported?

As an example, Figure 5 visualizes the spatial characteristics of the simulated and observed results on a map. However, these spatial visualizations are not at the core of our study, which is the aggregated statistics of SL and WSL. Thus, to save space and keep focus aggregated statistics, we would like to omit the map from the fsm_optim simulations.

Fig 6. What is "flat:2160", "flat:1940", etc for the three plots? Why 0% for lower altitude for S1? Is the patchiness? Please better explain this behavior.

Flat:XXXX is the elevation of the wet snow line in flat terrain. We included this in the caption.

Fig 8-9: it would be interesting using this plot to identify the possible sources of error operating on the different model parameters or equations. The fact that there is negative bias for both SL and WSL is interesting and should be better investigated.

We touched on this on the two last paragraphs of Sec. 4.2. In order to make the link clearer, we added a reference to the figure 8 (now Fig 9) at the beginning of the last paragraph (l. 355):

*Conversely, the wet snow line bias in our simulations is still dominated by a South-North contrast with fsm_optim (Fig. 9b)*

Dingman, S.: Physical hydrology, Waveland press, 2015

Douville H, Royer JF, Mahfouf JF. A new snow parameterization for the MeteoFrance climate model. Part I: validation in stand-alone experiments. Clim Dyn 1995;12:21–35.

Essery, R.: A factorial snowpack model (FSM 1.0), Geosci. Model Dev., 8, 3867–3876, https://doi.org/10.5194/gmd-8-3867-2015, 2015.

Goto, H., K. Kikuchi, and M. Kajikawa (2012). Influence of different surface soils on snow-water content and snow type of the snow cover. Japanese Journal of Snow and Ice 74:145–158

Premier, V., Marin, C., Bertoldi, G., Barella, R., Notarnicola, C., and Bruzzone, L.: Exploring the use of multi-source high-resolution satellite data for snow water equivalent reconstruction over mountainous catchments, The Cryosphere, 17, 2387–2407, https://doi.org/10.5194/tc-17-2387-2023, 2023.

Techel, F. and Pielmeier, C.: Point observations of liquid water content in wet snow – investigating methodical, spatial and temporal aspects, The Cryosphere, 5, 405–418, https://doi.org/10.5194/tc-5-405-2011, 2011.

References:

Fiddes, Joel, Kristoffer Aalstad, and Sebastian Westermann. "Hyper-Resolution Ensemble-Based Snow Reanalysis in Mountain Regions Using Clustering." Hydrology and Earth System Sciences 23, no. 11 (November 19, 2019): 4717–36. https://doi.org/10.5194/hess-23-4717-2019.

Cluzet, B. *et al.* Towards the assimilation of satellite reflectance into semi-distributed ensemble snowpack simulations. *Cold Regions Science and Technology* **170**, 102918 (2020).

Douville, H., Royer, J.-F. & Mahfouf, J.-F. A new snow parameterization for the Météo-France climate model: Part I: validation in stand-alone experiments. Climate Dynamics 12, 21–35 (1995).

Essery, R. A factorial snowpack model (FSM 1.0). Geosci. Model Dev. 8, 3867–3876 (2015).

Gardner, A. S. & Sharp, M. J. A review of snow and ice albedo and the development of a new physically based broadband albedo parametrization. J. Geophys. Res. 115, F01009 (2010).

Lamare, M. et al. Simulating Optical Top-Of-Atmosphere Radiance Satellite Images over Snow-Covered Rugged Terrain. (2020) doi:10.5194/tc-2020-104.

Largeron, C. et al. Toward Snow Cover Estimation in Mountainous Areas Using Modern Data Assimilation Methods: A Review. Frontiers in Earth Science 8, (2020).

Revuelto, J. *et al.* Assimilation of surface reflectance in snow simulations: Impact on bulk snow variables. *Journal of Hydrology* **603**, 126966 (2021).

---

## Author Comment (AC2)

Response letter for to first review round of:

**Using Sentinel-1 wet snow maps to inform fully-distributed physically-based snowpack models**

Bertrand Cluzet[1], Jan Magnusson[1], Louis Quéno[1], Giulia Mazzotti[2,1], Rebecca Mott[1], and Tobias Jonas[1]

[1]WSL Institute for Snow and Avalanche Research SLF, Davos, Switzerland

[2]Univ. Grenoble Alpes, Université de Toulouse, Météo-France, CNRS, CNRM, Centre d'Études de la Neige, Grenoble, France

In the following, comments from the reviewer appear in black. The authors' answer is in blue, with planned changes in italics.

Reviewer #2

I reviewed with interest the manuscript by Cluzet and colleagues on using S1 wet snow maps in snow model evaluation. Authors provided a novel contribution on the long-standing topic of using wet snow maps in snow modeling, by evaluating an operational snow model in Switzerland with several years of wet snow from S1 and fractional snow cover from S2. Results show a general good agreement, with a North vs. South bias that was improved by adjusting fresh snow albedo.

Overall, the research is novel, relevant, and timely. Particularly the intuition of a wet snow line is interesting and will likely be used in several future papers. Thus I am in favor of publication, after a minor revision.

Thank you for your positive feedback on our study and your valuable comments for improving the manuscript. Below, we have provided our responses to your comments and outlined the changes we plan to implement to enhance the paper.

Like R1, I was also a bit puzzled by the choice of adjusting fresh snow albedo as the main approach to correct the mismatch between model simulations and observations of wet snow. On the one hand, I understand that this variable is related to snowmelt onset and thus is one of the variables involved in this mismatch. On the other hand, in my understanding fresh snow albedo has a clear impact only when snow is fresh, while parameters related to the seasonal evolution of albedo seem more important to me here. Also the procedure that led to this adjustment is not clearly outlined and should be better discussed.

The fresh snow albedo affects the surface albedo not only after snowfall, but throughout the entire season. Therefore, it is more effective to adjust the start value (effective from the beginning) rather than the albedo decay functions (which only become effective after some time). The procedure that led us to this adjustment is now clearly explained in the response letter to review #1 (see above), and we performed significant changes to reflect that in Sec. 2.3 and in the discussion Sec. 4.2.

Another potential opportunity for improvement here is that the whole of the discussion around the wet snow data is performed in terms of wet snow line, rather than pixelwise values. Authors are clear on why they are doing so, and I generally agree. I still believe that computing confusion matrices for aspect or elevation classes would provide additional insights around model performance.

This is an interesting suggestion; however, computing confusion matrices may be misleading because the problem is slightly ill-posed. Intrinsically, the "wet snow" class is conditioned by the presence of snow. Computing a confusion matrix for wet snow would result in evaluating both "snow presence" and "wet snow" at once, with an unknown relative weight. Fig. 4 shows well that that Sentinel-1 wet snow retrieval relies on an auxiliary snow mask (see added elements in Section 2.2) to put pixels below 40% SCF in the "no snow our dry snow or patchy snow" whose accuracy is questionable at low elevations. In fig. 4, 6a and 7a, below 2000m, a significant number of pixels are classified by AlpSnow WSM as "no snow our dry snow or patchy snow" while the model still has snow. Arguably, we could filter the model with its own snow cover fraction, to reduce this mismatch, but we would nevertheless end-up evaluating this model component too, instead of just evaluating the wet snow part.

Some more minor comments:

- Line 26: maybe also mention lateral flow as a way for snowmelt to move away from the snowpack without exiting from the bottom of the local snow cover

We believe that this comment applied to paragraph 36-44, which was edited accordingly.

- Line 99: maybe better define what these hydrologic units are and how they were delineated?

The corresponding sentence was changed into:

*We focus our analysis on 16 subcatchments with relatively homogeneous snow conditions ("MEZ", as defined by the Swiss federal office of environment), which cover the alpine area of the domain.*

- Line 141: remove one "and"

Fixed

- Line 158: maybe spend some more words on this FSC = f(SWE, HS) relation here, given that it is quite important for this paper?

We agree and added further information including a reference to Helbig et al (2021) for full information:

*A subgrid parametrization is used to derive pixel-level snow cover fraction from seasonal values of SWE and HS (Helbig et al., 2021). This model component accounts for the impact of subpixel terrain roughness and slope variability on snow depth distribution at the peak of accumulation. For instance, smooth and flat pixels usually exhibit homogeneous subpixel SWE distribution, leading to a more rapid reduction in snow cover as SWE decreases Fresh snow events are assumed to produce abrupt yet transient increases in snow*

*cover fraction. Accounting for this is essential for comparing model outputs with satellite retrievals of snow cover fraction at this scale.*

- Section 3: I would recommend including more quantitative metrics here in place of wording like "excellent", "higher elevations", etc.

"Excellent" is used in l. 204 to qualify the comparison between flat-field snow depth observations and simulations since the curves are often barely distinguishable. Detailed performance statistics are available in Oberrauch et al (in press), Winstral et al., (2019) and Mott et al., (2023). We added a reference to these papers here:

*Overall, the match is excellent, in particular during the accumulation period, which is consistent with the fact that these observations were assimilated for correcting errors in the snowfall input (see Sec. 2.3) For a detailed evaluation of FSMoshd against station data, readers are referred to Winstral et al., 2019, Mott et al. 2023 and Oberrauch et al. (in press).*

"higher elevations" is used l.217 to qualify the difference between modelled and observed wet snow maps. The purpose of this paragraph is precisely to say that only qualitative information can be derived from map comparisons, stressing the need for the aggregation which follows, and gives quantitative numbers (e.g., l. 229). Nevertheless, to be more specific, we edited the instance l. 217:

*In the south-facing slopes, the observations indicate wet snow conditions several hundred meters higher than the model simulation.*

References:

1.

Helbig, N. *et al.* A seasonal algorithm of the snow-covered area fraction for mountainous terrain. *The Cryosphere* **15**, 4607–4624 (2021).
Oberrauch, M. et al. Improving fully distributed snowpack simulations by mapping perturbations of meteorological forcings inferred from particle filter assimilation of snow monitoring data, *Water Resources Research*, (in press)

---

## Author Comment (AC3)

General comments

Robust and original research that fits recent and growing interest for the snow cover in the field of hydrology due to variations and change of the climate. Please, see my comments below to improve the quality of your manuscript.

Thank you for your positive feedback on our study. Below, we have provided our responses to your comments and outlined the changes we plan to implement to enhance the paper.

Specific comments

Lines 1-2. "Sub-kilometric" and "large areas". Unclear the observation scale in your abstract. Please, revise it.

We don't think that it's necessary to be much more specific in the abstract, these terms are specific enough in the snow hydrology community.

Line 36. "Snow melt is not equivalent to snowmelt runoff". Please, explain better this concept in hydrology. Indeed, a large amount of snow can melt and recharge the groundwater bodies. Back-up the statement with recent literature from mountainous areas on snow melt aquifer recharge:

- Tracking flowpaths in a complex karst system through tracer test and hydrogeochemical monitoring: Implications for groundwater protection (Gran Sasso, Italy). Heliyon, 10(2), https://doi.org/10.1016/j.heliyon.2024.e24663

- Long-term trend of snow water equivalent in the Italian Alps. Journal of Hydrology, 614, 128532, https://doi.org/10.1016/j.jhydrol.2022.128532

In the context of snow modelling, snowmelt runoff is usually defined as the water flux that exits the snowpack at its base. So, with this statement we did not address any subsurface processes or catchment-scale considerations.

Line 88. I suggest to use the words "research questions" or "research objectives". Very good to be so clear when you explain the aim/objectives of your research. I see your good point!

The research questions will be removed in the revised manuscript (due to the feedback from our reviewers)

Lines 93-104. Please, provide basic information for your mountainous areas on the (i) climate, (ii) vegetation, and (iii) type of bedrock (fractured igneous-metamorphic rocks). All elements that affect infiltration and run-off of the melted snow.

As stated above, this study is about snowpack modelling, not about hydrological modelling. Therefore, subsurface hydrological processes are not considered. Nevertheless we will added glacier and vegetation outlines to Fig. 3.

Lines 217-218. Low and high elevations. Please, be more specific with regards to the topographic ranges.

The updated manuscript will be more specific about these terms.

Line 335. "This is not very informative". Please, insert the object after the word "this" to make the sentence clear.

"This" refers to the "information" in the previous sentence. We prefer to keep it like that to avoid the repetition of " information"/" informative".

Line 407. "Diversity of topographic conditions". Be more specific and not vague in your conclusions. I am trying to bring the impact out of your good research.

Specific information is available at the end of the conclusions, where the reader has already been familiarized with elevation, aspect, and slope. We don't think that we need to repeat ourselves here.

Lines 419-584. Please, integrate relevant literature on snow melt in hydrology, see above.

We included all the relevant literature from the list of papers provided.

Figures and tables

Figure 2. Provide explanation for the blue areas (0 observations per month) in the caption for the third figure in central-lower position.

In the meantime, we received additional remote sensing data, which is why we now also have data for NE Switzerland. The figure will be edited accordingly.

Figure 3. Dashed lines are better for the horizontal lines for elevations 2010 and 2290 mASL.

Thanks for the suggestion.

Figure 9. Letters and numbers on the axes are too small for all the four graphs.

This figure will be edited and should be more readable.